Learning from a provisioning site: code of conduct compliance and behaviour of whale sharks in Oslob, Cebu, Philippines

Schleimer Anna 1 2 anna.schleimer@odyssea.lu
Araujo Gonzalo 2
Penketh Luke 2
Heath Anna 2
McCoy Emer 2
Labaja Jessica 2
Lucey Anna 2
Ponzo Alessandro 2 3
1 Odyssea Marine Research and Awareness , Diekirch , Luxembourg
2 Large Marine Vertebrates Research Institute Philippines , Jagna, Bohol , Philippines
3 Large Marine Vertebrates Project Philippines, Physalus , Largo Callifonte, Roma , Italy
Kramer Donald
Electronic publication date: 2015 Nov 26
Publication date: 2015
Volume: 3
Electronic Location ID: e1452
Received 2015 Sep 9; Accepted 2015 Nov 6
Copyright: © 2015 Schleimer et al.
Copyright year: 2015
Copyright holder: Schleimer et al.
License: This is an open access article distributed under the terms of the Creative Commons Attribution License, which permits unrestricted use, distribution, reproduction and adaptation in any medium and for any purpose provided that it is properly attributed. For attribution, the original author(s), title, publication source (PeerJ) and either DOI or URL of the article must be cited.
License URL: https://creativecommons.org/licenses/by/4.0/

Keywords: GLMM, Conditioning, Whale shark, Oslob, Learning, Precautionary principle, Shark-based tourism, Provisioning, Behaviour, GEE

Funding: M5-SRL Roma M5-SRL Roma provided funding for completion of this study. The funders had no role in study design, data collection and analysis, decision to publish, or preparation of the manuscript.

==============================
While shark-based tourism is a rapidly growing global industry, there is ongoing controversy about the effects of provisioning on the target species. This study investigated the effect of feeding on whale sharks (Rhincodon typus) at a provisioning site in Oslob, Cebu, in terms of arrival time, avoidance and feeding behaviour using photo-identification and focal follows. Additionally, compliance to the code of conduct in place was monitored to assess tourism pressure on the whale sharks. Newly identified sharks gradually arrived earlier to the provisioning site after their initial sighting, indicating that the animals learn to associate the site with food rewards. Whale sharks with a long resighting history showed anticipatory behaviour and were recorded at the site on average 5 min after the arrival of feeder boats. Results from a generalised linear mixed model indicated that animals with a longer resighting history were less likely to show avoidance behaviour to touches or boat contact. Similarly, sequential data on feeding behaviour was modelled using a generalised estimating equations approach, which suggested that experienced whale sharks were more likely to display vertical feeding behaviour. It was proposed that the continuous source of food provides a strong incentive for the modification of behaviours, i.e., learning, through conditioning. Whale sharks are large opportunistic filter feeders in a mainly oligotrophic environment, where the ability to use novel food sources by modifying their behaviour could be of great advantage. Non-compliance to the code of conduct in terms of minimum distance to the shark (2 m) increased from 79% in 2012 to 97% in 2014, suggesting a high tourism pressure on the whale sharks in Oslob. The long-term effects of the observed behavioural modifications along with the high tourism pressure remain unknown. However, management plans are traditionally based on the precautionary principle, which aims to take preventive actions even if data on cause and effect are still inconclusive. Hence, an improved enforcement of the code of conduct coupled with a reduction in the conditioning of the whale sharks through provisioning were proposed to minimise the impacts on whale sharks in Oslob.

Introduction

Shark-based tourism is a rapidly growing global industry (Gallagher & Hammerschlag, 2011). With its economic benefits extending throughout the community, shark-based tourism has allowed some communities to improve their economic status (Quiros, 2007; Brunnschweiler, 2010; Gallagher & Hammerschlag, 2011; Vianna et al., 2012). Despite numerous advantages of managing shark species as a non-consumptive resource for both the target species and local communities, the appropriateness of the term ‘non-consumptive’ has been questioned in recent years (Gallagher et al., 2015). Initially a non-consumptive activity relating to wildlife recreation was defined as “human recreational engagement with wildlife wherein the focal animal is not purposefully removed or permanently affected by the engagement” (Duffus & Dearden, 1993). The potential effects of shark-based tourism on the focal species and its ecosystem have been the topic of considerable debate, especially when provisioning through chumming, baiting or feeding occurs (Orams, 2002; Gallagher et al., 2015; Brena et al., in press). Proposed impacts of provisioning in elasmobranchs include direct impacts on the physiology or behaviour of the provisioned animals as well as indirect effects, cascading through the ecosystem by altering predator–prey interactions through changes in predator abundance or composition at the feeding site (Orams, 2002; Brunnschweiler & Baensch, 2011; Brunnschweiler, Abrantes & Barnett, 2014). However, while direct effects of provisioning were reported for some elasmobranch species (Semeniuk & Rothley, 2008; Semeniuk et al., 2009; Clua et al., 2010; Fitzpatrick et al., 2011), other studies found no evidence for behaviourally mediated effects of provisioning at large spatial and temporal scales (Hammerschlag et al., 2012). Furthermore, lack of data precludes determination of ecosystem-wide impacts mediated by provisioning (Burgin & Hardiman, 2015). Maljković & Côté (2011) suggested that the amount of food provided to Caribbean reef sharks (Carcharhinus perezi) was insufficient to meet their daily requirement and that provisioning was unlikely to alter their role as apex predators in the ecosystem. Despite considerable research in the recent years, no general trend about the reported effects of provisioning on shark species seems to be obvious; in contrast, effects appear to be species, context and site dependent and generalisations across provisioning sites may result in a distorted image of the true effects (Brunnschweiler & Barnett, 2013; Gallagher et al., 2015; Burgin & Hardiman, 2015; Brena et al., in press).

However, while the scientific community has yet to come to a consensus about the effects of shark-based tourism on the target species, it has been acknowledged that feeding sites can offer a unique opportunity to study the behaviour of these otherwise elusive species (Hammerschlag et al., 2012). The unsuitability of studying large elasmobranch species in a captive environment, and time-consuming and costly field experiments, have resulted in a lack of knowledge on cognitive abilities in elasmobranchs compared to other vertebrates (Guttridge et al., 2009; Schluessel, 2015). The direct response of sharks to tourism is commonly assessed using an ethological approach, based on direct observations of behaviours (Pierce et al., 2010; Guttridge & Brown, 2014; Haskell et al., 2014). By creating a descriptive catalogue of behaviours, the so-called ethogram, the occurrence of various behaviours can be assessed in relation to certain stimuli (Gruber & Myrberg, 1977). Provisioning sites can offer frequent and predictable encounters with the target species, as well as close interactions, and pre-existing infrastructures. However, the artificial nature of such sites should be taken into account because behaviours occurring at these sites may not be representative of natural behaviours or grouping patterns (Semeniuk & Rothley, 2008).

Additionally, sharks are often confronted with completely novel situations and stimuli at provisioning sites, offering a valuable insight into the learning capacities of these species (Guttridge et al., 2009). Learning is defined as the adaptive modification of behaviour based on experience, allowing an animal to develop, within its lifetime, an adaptive response to a novel situation that has potentially never been encountered in the species’ evolutionary past (Kawecki, 2010; Gallagher et al., 2015). In the case of repeated exposure to stimuli, learning can occur either through learning an association or relationship between two events, known as associative learning (e.g., conditioning or observational learning) or by non-associative learning, where a single stimulus leads to habituation or sensitisation (Lieberman, 2000). Results from previous studies indicate that the cognitive abilities in elasmobranchs are as well developed as in teleosts or other vertebrates (Schluessel, 2015). Their ability to learn, habituate, and remember spatial maps was suggested to facilitate efficient foraging, predator avoidance, habitat selection and mate choice in a dynamic environment (Guttridge et al., 2009; Schluessel, 2015). However, many unanswered questions still remain about basic principles of shark learning, notably whether any intra- or interspecific differences in learning exist, where shark learning capabilities fit in with other vertebrates, and how long sharks can memorise associations (Guttridge et al., 2009).

In this study, we used the setting of a provisioning site to gain insight into the learning capacity of the whale shark (Rhincodon typus). We hypothesised that learning plays an important role when assessing the impact of tourism on elasmobranchs and that the animals’ capacity to learn (e.g., through conditioning and habituation) needs to be taken into account when developing regulations for human-shark interactions. Current management plans of shark tourism rely on compliance to codes of conduct, which have been implemented by the majority of shark tourism operators as a precautionary procedure to mitigate potential negative impacts on sharks and guests (Richards et al., 2015). Codes of conduct aim to regulate human behaviour during shark interactions, in terms of maximum people per shark, minimum distance to the shark, and general practices such as no touching or flash photography (Richards et al., 2015). However, despite the observation of widely site- and species-specific effects of nature-based tourism on sharks, only some codes of conduct are the direct product of site-specific scientific studies (e.g. Pierce et al., 2010). Additionally, poor enforcement and impracticality may lead to low compliance to these codes of conduct (Quiros, 2007; Smith, Scarr & Scarpaci, 2010).

This study investigated the adaptive behavioural response of whale sharks to novel stimuli at a provisioning site, with particular emphasis on learning processes over time and the potential importance of learning when developing codes of conduct. Firstly, changes in tolerance levels to perceived disturbance at the provisioning site were evaluated. The setting allowed sequential sampling of the same individuals exposed to different levels of disturbance, hence allowing to detect potential habituation or sensitisation processes (Bejder et al., 2009). Secondly, we examined the occurrence of associative learning in the provisioned whale sharks. The repeated exposure to food rewards was expected to lead to conditioning in sharks through positive reinforcement when the correct voluntary behaviour was displayed. Specifically, changes in feeding behaviour as well as time of arrival to the provisioning site were investigated over time. Compliance to a code of conduct was considered to act as a proxy for tourism pressure (Quiros, 2007) and was discussed within the context of the behavioural modifications observed in the whale sharks.

Methods

The methods presented here were conducted in accordance with national and local laws in respect of animal welfare. The Bureau of Fisheries and Aquatic Resources—Region 7, issued the authors a Gratuitous Permit (NO.01-2013), and a Memorandum of Agreement was signed with the Department of Environment and Natural Resources and DA-BFAR7. The Municipality of Oslob granted the authors a Prior Informed Consent document.

Study site

The study site was located in the coastal waters off Barangay Tan-Awan, Municipality of Oslob, Cebu Province, Philippines. The provisioning of whale sharks occurs within an interaction area of about 65,000 m2, located about 50 m from shore, as described and illustrated by Araujo et al. (2014). The municipal government has granted permission to provision whale sharks only to members of the Tan-Awan Oslob Whale Shark Fishermen Association (TOWSFA). Year-round, feeders on one-man paddleboats provision the animals from 6 am to 1 pm, using up to 150 kg of food daily, depending on shark and tourist numbers. The food consists mainly of sergestid shrimp, which is locally known as uyap. Same as the provisioning, the interaction with whale sharks is only allowed between 6 am and 1 pm and no whale shark watcher is allowed in the interaction area outside those times. Tourists receive a briefing on the interaction guidelines prior to their whale shark watching and are taken to the interaction area on paddleboats of approx. 5 m, carrying up to 12 people for 30 min of whale shark watching. Once the tourist boats are aligned by tying to neighbouring boats, tourists can choose to enter the water and either hold on to the boat or snorkel with the whale sharks. Feeders lure the whale sharks with food along the line of tourist boats. On days without current, tourist boat formation may be of a more random nature and circles of boats around sharks have been observed. Scuba divers already on boats when arriving to the interaction area are not required to pass through the briefing centre on land; instead, the ordinance states that the orientation should be conducted onboard. Scuba divers are dropped from the motor boats moored outside the interaction area for an interaction of up to an hour.

Code of conduct

The code of conduct for the whale shark activities taking place in the municipal waters of Oslob, Cebu, was approved as a municipal ordinance on 6 January 2012, and further amended on 12 April 2012. The ordinance states that whale shark watching is limited to a thirty-minute interaction. Tourists are not allowed to stay within 5 m directly beside or behind the caudal fin and within 2 m in front of the whale shark. Additionally, a maximum of six whale shark watchers (i.e., snorkelers or boat-holders) and four scuba divers per whale shark are permitted at any one time. Heavy splashing, riding, touching using hands, feet, camera or pointers are strictly prohibited. Compliance with the code of conduct was assessed by investigating the number of instances in which swimmers, boat-holders, and scuba divers were observed within 2 m of the whale shark during focal follows. While tourists are supposed to keep a 5 m distance from the flanks and caudal fin of the whale shark, the 2 m limit, corresponding to about a human body length, was considered a definite infraction of the code of conduct and excludes the people that are just marginally breaking the code of conduct. The total number of people within 10 m of the whale shark was used to assess the compliance with the recommended number of people per shark. While the ordinance states the maximum number of people per shark, it does not explicitly describe the distance at which an observer is considered interacting with the shark. Instead the 10 m limit was chosen based on the feasibility of counting the number of people accurately with varying degrees of visibility during the seasons.

Behavioural surveys

Researchers conducted in-water surveys with the aim to record both behaviour of whale sharks and compliance of tourists and boatmen to the code of conduct. Each in-water survey lasted for 20 min during which a single shark was followed. Focal follows are a commonly used field method to sample behaviours of readily identifiable individuals or groups (Altmann, 1974; Mann & Würsig, 2014). Upon entering the interaction area, the first shark seen was chosen as the focal follow shark to ensure random sampling of animals. Data collected during each 20-minute segment included tourist numbers, shark behaviour, and other events, such as shark contact. Data on tourist numbers were collected every 5 min and consisted of snorkelers and boat-holders within 2 m of the shark, total number of tourists (snorkelers and boat-holders) within 10 m of the shark, and scuba divers within 2 m and 10 m of the shark. Additionally the number of other sharks within 10 m was recorded. The predominant behaviour of the shark was recorded every 5 min based on predefined categories: horizontal feeding (HF), vertical feeding (VF), free swimming (FS), or natural feeding (NF; Table 1). Additionally active and passive guest contacts, boat contacts and shark-to-shark contacts were recorded with the subsequent reactions as they happened (Table 1). Reactions to such events were categorised as no response, swimming away, circling behaviour, switching feeder boats, and thrashing of the body (Table 1). If a shark swam out of sight during a focal follow, the session was ended if the shark was not relocated within 3 min.

Table 1 Definitions of the predominant behaviour as recorded on a 5 min basis during focal follows of whale sharks, as well as definitions of events and subsequent reactions.

	Definition	
Predominant behaviour		
Horizontal feeding	Shark actively swims behind feeder boats with its body angled horizontally (variation of angle depending on speed of current and feeder boats).	
Vertical feeding	Shark is in a stationary position, with its body in a vertical orientation with its mouth just below the water surface. Food is ingested by gulping water using a suction technique.	
Natural feeding	Shark swims with either partially or totally open mouth displaying passive or active feeding in an area away from the feeder boats.	
Free swimming	Shark swims with mouth closed, independently of feeder boat proximity.	
Events		
Active touch	Guest intentionally approaches the animal and initiates shark contact with any body part or gear (e.g., fins, camera, camera pole).	
Passive touch	Any contact between shark and guest where the guest does not intentionally approach the animal.	
Feeder contact	Feeder intentionally touches the shark with any body part or his paddle.	
Shark-to-shark contact	Two or more sharks make physical contact.	
Boat contact	Shark and any boat in the interaction area make physical contact.	
Reactions		
No reaction	No evident behavioural change recorded immediately after the observed event.	
Swam off	Shark changes behaviour abruptly and swims away without depth variation.	
Dive	Shark changes behaviour and descents to greater water depths.	
Bank	Shark rolls and orientates its dorsal side towards the perceived threat.	
Cough	Shark forcefully expels water and other material out of the mouth.	
Eye roll	Shark rotates eye backward into the eye socket.	
Violent shudder	Shark physically shakes its body.	

Each focal follow shark was identified using photo-identification (photo-ID) of the unique spot pattern behind the gill slits and above the pectoral fin on the left side of the body. Photo-ID on whale sharks has been successfully used for the study of residence patterns (Araujo et al., 2014), to estimate survival (Bradshaw, Mollet & Meekan, 2007), and to investigate scarring patterns and mortality rates (Speed et al., 2008). Additionally, animals were sexed by the absence (females) or presence (males) of pelvic claspers (Araujo et al., 2014). The size of the shark was visually estimated by the researcher, a method which has been previously validated using measurements from laser photogrammetry for comparison (Araujo et al., 2014).

Upon arrival to the study site, the time of the first feeder boats as well as first shark observed from an outlook on land were noted. Binoculars were used to either spot dorsal or caudal fins or the shadow of a shark under the water surface. When the researcher entered the water at 6 am by snorkelling out to the interaction area from shore, photo-ID pictures of the first shark sighted in water were recorded along with the time of this first sighting. These data were used to assess when sharks arrived at the study site relative to the onset of feeding and to test whether past sighting history affected time of arrival. Additionally, shark presence was recorded during every hourly photo-ID session from 6 am to 1 pm from February 2014 onwards (see Araujo et al., 2014 for details). Photo-ID pictures allowed the identification of individual sharks or the assignment of a new identification number in case the whale shark was not found in the ongoing photo-ID catalogue by two experienced researchers. In addition to the local identification number, pictures of each shark were uploaded to the international online whale shark database ‘Wildbook’ on www.whaleshark.org, where an international identification number was assigned to each individual. The number of session in which a previously unidentified shark was observed was extracted from the photo-ID dataset to assess whether new sharks were more likely to be sighted at a certain time of day.

Data analysis

A binomial Generalised Linear Mixed Model (GLMM) was fitted to investigate which factors affect avoidance behaviour. GLMMs are a combination of linear mixed models, allowing the incorporation of random effects, and generalised linear models, which apply link functions and exponential families to deal with non-normal data (Bolker, Brooks & Clark, 2009). Avoidance behaviour was coded as a binary response, where no reaction to event was reported as 0 and behaviours such as swimming off, banking, eye roll, diving and change in direction were marked as 1. Potential explanatory variables included the event (Table 1), predominant behaviour, number of previous sightings (i.e., number of confirmed daily sightings of the individual shark), sex, estimated size, current, visibility, and number of swimmers within 10 m of the shark. Current was measured on a scale of 1–4 (1 = no current, 2 = researcher requires effort to swim, 3 = researcher experiences difficulty to swim against current, 4 = researcher cannot swim against current). Visibility was recorded as the maximum distance in meters from which a researcher could confidently identify a whale shark from the spot patterns. Due to problems of model convergence, the variable ‘previous number of visits’ was rescaled by dividing through the interquartile range (Babyak, 2009). Shark ID was added as a random effect to avoid pseudoreplication, which occurs when replicates are not statistically independent (Hurlbert, 1984; Dawson & Lusseau, 2005). Free swimming and passive touch were specified as the reference baseline levels, because sharks were expected to be less likely to show avoidance behaviour while feeding and in response to minor disturbance. Model selection was based on single term deletion, which drops one explanatory variable, in turn, and each time applies an analysis of deviance test (Zuur et al., 2009). GLMM parameters were estimated using Laplace approximations. Models were refitted until all explanatory variables were significant with p < 0.05 using the lme4 package in R (Bates et al., 2014).

While errors were assumed to be independent for the avoidance behaviour data, which were logged as they occurred throughout the focal follow, data on predominant behaviour were collected systematically every 5 min during focal follows and were therefore prone to temporal autocorrelation. Falsely assuming independence can lead to invalid significance tests, because standard errors and p-values tend to be too small and irrelevant explanatory variables may be falsely kept in the models (Panigada et al., 2008; Bailey et al., 2013). Generalised estimating equations (GEEs) account for temporal autocorrelation and were considered the most appropriate models to analyse behavioural data collected during repeated focal follows (Liang & Zeger, 1986; Bailey et al., 2013). More specifically, the unrealistic assumption of constant error variance is replaced with a correlation structure in GEEs. In this case, errors were allowed to be correlated within data collected from the same individual. The occurrence of vertical feeding was modelled using a marginal GEE approach with a binomial error distribution and logit link function. Shark ID was defined as a block of data, within which observations were temporally autocorrelated, using the auto-regressive correlation structure. GEEs yield consistent estimates even with misspecification of the working correlation matrix (Liang & Zeger, 1986). Explanatory variables that were considered for inclusion into the model included number of previous visits, visibility, current, whale shark sex and size. GEEs were fitted using the package geepack version 3.1.3. (Yan, 2002; Højsgaard, Halekoh & Yan, 2006) in R version 3.1.1 (R Core Team, 2014). Model selection was based on backward selection using analysis of deviance tests (Zuur et al., 2009).

In order to investigate whether the first shark recorded every morning at the onset of feeding had a longer resight history compared to sharks never reported as first shark of the day, a Welch two sample t-test was run to compare the number of visits between both groups i.e., first sharks and non-first sharks. A Chi-square test was used to compare the total number of newly identified sharks spotted in each hourly photo-ID session (6–7, 7–8, 8–9, 9–10, 10–11, 11–12, 12–13). Additionally, new sharks that had over 10 resightings within 20 days of their first sighting at the study site, and therefore showing repeated visitation at the study site, were chosen to investigate changes in time of arrival. For all statistical analyses the level of significance was set at p < 0.05.

Results

Data summary

From 31 March 2012 to 20 June 2014, a total of 171 individual whale sharks were identified, of which 59 sharks were followed, leading to 1,109 focal follow surveys. These surveys typically lasted 20 min, but on 27 occasions, surveys were terminated prematurely when the whale shark swam out of sight and was not resighted within the following 3 min. These 27 surveys were excluded from further compliance analysis leaving data from 672 focal follows in 2012, 291 in 2013 and 119 in 2014. The remaining 1,082 complete focal follow surveys added up to circa 357 h worth of data. Some individuals showed very high site fidelity to the feeding site, allowing sequential sampling of the same individuals over a prolonged time period, with 18 individuals being subject to focal follows in at least two consecutive years. A full description of resight histories and visitation patterns are given by Araujo et al. (2014).

Compliance

Compliance to the required minimum distance from the shark was investigated during the 1,082 complete focal follows. During 907 focal follows (84%) at least one snorkeler or boat-holder was observed being less than 2 m away from the whale shark, inconsistent with the code of conduct (Table 2). On average there were 1.9 snorkelers or boat-holders (S.D. = 2.6) and 0.2 scuba divers (S.D. = 0.9) within 2 m of the whale shark recorded every 5 min. The maximum number of snorkelers and boat-holders around a single shark was observed on 7 September 2013 when 19 people were recorded within 2 m of a whale shark and, respectively, on 10 December 2012 when 10 scuba divers were closer than 2 m from the shark.

Table 2 Seasonal percentage non-compliance in terms of minimum distance kept to the whale shark and active touches by guests.

Regulation	Percentage non-compliance	
	2012 season	2013 season	2014 season	
Snorkeler closer than 2 m from shark	65.6%	na	84.9%	
Boat-holder closer than 2 m from shark	54.3%	na	76.5%	
Diver keeping closer than 2 m from shark	26.3%	na	20.2%	
Snorkeler and boat-holder closer than 2 m from shark	78.6%	90.7%	96.6%	
Active touches by guests	8.7%	31.4%	14.3%	

Data on the number of guests within 10 m of sharks was missing from 62 focal follows in 2012, the remaining 1,020 focal follows showed that in 56.1% of the surveys the maximum number of snorkelers allowed per shark (i.e., 6) was exceeded (Fig. 1). The maximum number of snorkelers and boat-holders within 10 m of a shark was 33. Records of the number of scuba divers within 10 m of a shark only started in July 2013. During these 406 focal follows, the maximum of 4 scuba divers was exceeded on 79 occasions (19.5%). Twenty-one scuba divers within 10 m of a shark was the highest number of divers recorded per shark.

Figure 1 Frequency distribution of maximum number of snorkelers and boat-holders recorded within 10 m of the whale shark per focal follow.

The red line shows the maximum number of people allowed per shark.

From May 2012 to January 2013, researchers systematically counted the amount of active touches from guests and feeders on sharks. A total of 4,832 active touches were recorded over 545 focal follows. Feeders pushing away sharks with their feet or petting the sharks with their hands accounted for the majority of these active contacts (97.6%); while guests were observed touching sharks 117 times. During the 2013 and 2014 survey seasons, feeder touches were no longer systematically counted, because of the lack of reaction observed in the whale sharks. On 114 occasions guests were recorded to actively touch whale sharks during this same period.

Avoidance behaviour

On 232 occasions since 2012, events such as active touches, boat contact, shark-to-shark contact and passive touches were recorded with the corresponding response of the shark (reaction/no reaction). The occurrence of avoidance behaviour was significantly influenced by the number of the shark’s previous visits (χ2(1, N = 232) = 4.31, p = 0.038), the behaviour at the time of the event (χ2(2, N = 232) = 16.26, p < 0.001) and the event that caused the response (χ2(4, N = 232) = 25.79, p < 0.001). Sharks were less likely to show avoidance when they were vertical or horizontal feeding and had a long history of sightings (Table 3 and Fig. 2). The odds of observing a behavioural response increased when sharks were free swimming and had a short history of sightings. The event that elicited the strongest response was shark-to-shark contact, while passive contact had little effect.

Figure 2 GLMM results of the occurrence of avoidance behaviour in relation to the number of previous daily sightings.

(A) Fitted values of GLMM show overall effect of the number of previous visits on the probability of avoidance behaviour with line of best fit; (B) predicted probability of avoidance behaviour after shark-to-shark contact of a whale shark that is feeding vertically. Shaded area shows 95% confidence interval.

Table 3 Parameter estimates of GLMM on avoidance behaviour with standard errors (s.e.).

Fixed effect	Coefficient	s.e.	
Intercept	−0.21	0.74	
Number of previous visits	−0.82	0.43	
Behaviour: horizontal feeding	−1.29	0.56	
Behaviour: vertical feeding	−1.69	0.44	
Event: boat contact	0.46	0.71	
Event: feeder contact	0.22	0.91	
Event: active touch	1.45	0.81	
Event: shark-to-shark contact	2.18	0.64	

Effect of provisioning on feeding pattern

A total of 357 h of focal follow observations were collected on 59 different sharks. The main behaviour observed in the whale sharks at the provisioning site was vertical feeding (58.2%), followed by horizontal feeding (21.8%) and free swimming (20.2%), while natural feeding only accounted for 0.02% of all observations. The GEE indicated that the number of previous visits, as well as the current were significant predictors of observing vertical feeding (Table 4 and Fig. 3). The estimated autocorrelation parameter of 0.41 (SE = 0.03) indicates moderate autocorrelation, and justified our use of the GEE approach.

Figure 3 Predicted values of vertical feeding occurrence from GEE model against number of previous daily sightings with 95% confidence intervals.

Table 4 Results from the generalised estimating equation model with autoregressive correlation structure on the occurrence of vertical feeding.

Parameter	Coefficient estimate	Standard error	Wald	p	
Intercept	0.378	0.188	4.04	0.044	
Previous visits	0.001	0.0004	13.95	<0.001*	
Current	−0.200	0.080	6.16	0.013*	
Notes.

* indicates significance at p < 0.05 level.

Trends in shark arrival

Records of shark session presence during each individual 1 h session from 6 am to 1 pm started in February 2014, since when 51 new sharks were identified and added to the photo identification catalogue. These new sharks were more likely to be encountered during the later survey sessions, with the highest number of new sharks (N = 16) identified in the last session from 12 to 1 pm (χ2(6) = 16.67, p = 0.01; Fig. 4). Only 3 new sharks were identified during the very first session from 6 to 7 am. However, one of these sharks was anecdotally identified through photo-ID in the area prior to the start of our records. After their initial sighting, newly identified sharks tend to arrive at the interaction area progressively earlier (Fig. 5). During the first session from 6 to 7 am sharks were spotted from land on average 5 min after the feeder boats entered the interaction area. The first sharks to be identified in water every morning had a statistically significantly longer history of sightings compared to sharks that have never been reported as first sharks (T31.25 = 6.15, p < 0.001). The average number of visits of “first” sharks was 328.5 compared to 19.1 visits of sharks never observed as first shark.

Figure 4 Total number of previously unidentified sharks encountered in each hourly survey session from 6 am to 1 pm.

Figure 5 Arrival time to the study area of 11 sharks on the 10 sightings following their first appearance in the interaction area.

Lines represent best fit lines for each individual. The trend was negative in all cases and statistically significant in 6 out of 11 cases (statistical tests not shown).

Discussion

This study highlights the effects of provisioning on the feeding and avoidance behaviour of whale sharks, while using the particular setting of a provisioning site to gain an insight into learning capacities of whale sharks. Results provide evidence for both associative learning, in terms of associating the study site with food and adapting their feeding behaviour, and habituation, in terms of increasing tolerance levels in response to proximity of people and other sharks.

At the time of writing, 821 individuals were identified on “Wildbook for Whale Sharks” in the Philippines, with natural aggregations occurring in Donsol, Sorsogon Province, and Pintuyan, Southern Leyte, but sightings are reported country-wide (Quiros, 2007; Araujo et al., 2014). A long history of fishery and trade of whale sharks in the Bohol Sea and satellite telemetry data confirm their natural occurrence in these waters (Alava et al., 2002; Eckert et al., 2002). However, it remains unknown how individuals recruit into the artificial feeding aggregation at the provisioning site in Oslob and how many sharks frequented the area before the beginning of the provisioning activities (Araujo et al., 2014). Our observations that new whale sharks are mainly identified in the later survey sessions, when the waters are already saturated with uyap, could be an indication that whale sharks rely on chemosensory cues to detect nutrient rich waters. Aquarium-held whale sharks responded to chemical stimuli cues, such as dimethylsulfide (DMS) and homogenised krill, by exhibiting pronounced ingestive and search behaviours, suggesting that olfactory cues may play an important role to locate food in an environment with sparsely distributed resources (Dove, 2015). The highly developed olfactory epithelium located in the nares and corresponding processing centre in the brain further support this hypothesis (Dove, 2015). The use of chemosensory cues to locate patchily distributed prey over large spatial scales has also been proposed for other wide-ranging marine predators such as the basking shark (Cetorhinus maximus; Sims and Quayle 1998), the wandering albatross (Diomedea exulans; Nevitt, Losekoot & Weimerskirch, 2008), and the bowhead whale (Balaena mysticetus; Thewissen et al., 2011). Currents leading away from the provisioning site could disperse chemosensory cues over a large area and attract new whale sharks which were moving through the area. Of the 3 sharks that were newly identified in the first session, at least one had been sighted at the provisioning site prior the start of our surveys. While provisioning presumably started in September or October 2011, dedicated photo-ID surveys did not start until March 2012, which means that individuals could have been wrongly classified as new sharks.

Once new whale sharks were sighted in the study area, their time of arrival to the study site shifted to earlier sessions within 10 resightings for the 11 sharks studied. These observations were consistent with reports of shorter time of arrival of white sharks (Carcharodon carcharias) to a chumming site in South Africa with increased experience (Johnson & Kock, 2006) and anticipation on feeding times in pink whiprays (Himantura fai) in French Polynesia (Gaspar, Chateau & Galzin, 2008). While our results are not conclusive, they suggest that after the initial discovery of this new feeding site, the sharks may no longer solely rely on the presence of chemosensory cues in the water, but learn and remember the association of this particular site with the presence of food. For example, by creating a cognitive spatial map, sharks may be able to return to the provisioning site even in the absence of chemosensory cues. This would explain why whale sharks with a long resight history showed anticipatory behaviour by appearing at the provisioning site only minutes after the feeder boats entered the area and before provisioning had started. The ability to create spatial maps has been recognised in several elasmobranchs species, including freshwater stingrays (Potamotrygon motoro), bamboo sharks (Chiloscyllium punctatum and C. griseum), Port Jackson sharks (Heterodontus portusjacksoni) and lemon sharks (Negaprion brevirostris) (Edren & Gruber, 2005; Schluessel & Bleckmann, 2005; Papastamatiou et al., 2011; Schluessel & Bleckmann, 2012; Guttridge & Brown, 2014). As other large planktivores, whale sharks face a highly dynamic and rapidly changing environment. Sims et al. (2006) suggested that learned responses to previously encountered prey distributions might explain the highly efficient foraging pattern observed in basking sharks.

The main behaviours observed in the interaction area were vertical feeding, followed by horizontal feeding and free swimming. The GEE model indicated that whale sharks were less likely to display vertically feeding with a stronger water current. When the current was strong, whale sharks had to actively swim to counter-act the effect of the current and to maintain their position around the feeder boat, whereas no such energy expense was required in the absence of current, which promoted stationary, vertical feeding. Interestingly, the model showed that the probability of vertical feeding increased with the shark’s experience, which is consistent with observations of new individuals mainly displaying passive feeding or surface feeding away from the feeder boats. Nelson & Eckert (2007) described vertical feeding as part of the natural repertoire of feeding behaviours in whale sharks and their observations suggested that prey abundance may influence which feeding technique was used. The mean zooplankton density during vertical feeding was only a quarter of that during active feeding (Nelson & Eckert, 2007). At the provisioning site, only small handfuls of food are provided to lure the shark along the tourist boats. So while the waters are saturated with chemical stimulus plumes released from the uyap, which have been reported to trigger ingestion behaviours in whale sharks (Dove, 2015), the actual density of prey in the water is very low. We hypothesise that the increase in vertical feeding in experienced sharks at this provisioning site is a learnt behaviour as a result of conditioning through positive reinforcement. The continuous food rewards offer a strong incentive for the whale sharks to display the wanted behaviours (i.e., trailing behind the feeder boats) and the animals might quickly learn to associate the feeder boats with food to increase their foraging efficiency. Of the three filter feeding shark species, whale sharks are the only species to be able to suction filter feed, or to feed vertically, which requires no forward movement (Nelson & Eckert, 2007). It was proposed that this additional foraging technique may allow the whale shark to compensate for its less efficient filter-feeding apparatus compared to that of the basking and megamouth shark (Heyman et al., 2001; Nelson & Eckert, 2007). Whale sharks are known to have a very plastic feeding behaviour and to opportunistically feed on a variety of prey species and sizes, an ability which may allow them to optimise foraging in largely nutrient poor tropical waters (Heyman et al., 2001; Rohner et al., 2015). Feeding techniques such as vertical feeding require a high level of coordination, which might explain the large and highly foliated cerebellum in whale sharks, indeed one of the largest cerebellums within the chondrichthyan clade, which is thought to be involved in modulation of motor programs, self-motion error correction, and dynamic state estimation for coordination of target tracking (Yopak & Frank, 2009).

It should be noted that by choosing the first shark the researcher encountered as focal follow shark, free swimming whale sharks at the deeper, more offshore edge of the interaction area were less likely to be included in the data set. Observations were therefore biased towards sharks with high visitation rates. However, the bias that was potentially introduced through the sampling regime was not expected to have an effect on the interpretation of results as our primary interests were sharks that regularly returned to the provisioning site and interacted with the feeders and tourists.

In addition to modification of feeding behaviour through associative learning, the GLMM provided evidence for changes in the avoidance behaviour of the whale sharks with longer re-sighting histories. Experienced sharks were less likely to show responsive behaviour to actions such as active or passive human contacts and shark-to-shark contact, indicating that the animals became habituated over time. Habituation was defined as a reduction of response to a repeated stimulus, as individuals learn that there are neither adverse nor beneficial consequences of the occurrence of the stimulus (Humphrey, 1933). Organisms are constantly exposed to a myriad of sensations and habituation allows them to filter out irrelevant stimuli and to focus on important ones (Rankin et al., 2009). By habituating to physical contacts with humans and other sharks, whale sharks can maximise time spent feeding. In this case, the response to the stimulus was often followed by a food reward, which is why the observed increase in tolerance levels may partially be the result of conditioning, rather than habituation in its pure sense (Bejder et al., 2009). Sharks were most likely to show a response when they were free swimming, in which case their behaviour was not reinforced by provision of food, while they were least likely to show avoidance while they were vertically feeding. In terms of energetic costs, breaking out of the vertical position to swim off is likely more costly than showing avoidance while already swimming freely. The stimulus that elicited the strongest response was shark-to-shark contact, which often resulted in sharks violently thrashing and swimming off. The underlying cause for this strong reaction is not known and could be related to social hierarchy, aggression, or discomfort of physical contact. However, the population in Oslob was described as a male-dominated aggregation of juvenile whale sharks (Araujo et al., 2014) and neither size nor sex were significant predictors of avoidance behaviour. R. typus is described as a far-ranging, solitary species, which forms almost sex-segregated seasonal aggregations in connection to specific productivity events (Heyman et al., 2001; Rowat & Brooks, 2012). As solitary animals, direct body contact might just not be part of their daily social behaviour, but observations on the occurrence of intra-specific body contacts are required to fully understand the strong responsive behaviour. The second strongest stimulus to cause a behavioural response was active touches by guests, such as grabbing the dorsal or caudal fin. In contrast, accidental touches with guests and contacts with boats and the feeder did not elicit a strong behavioural response. Quiros (2007) found that active touches by people were a significant predictor of the occurrence of violent shuddering in whale shark during free swimming encounters in Donsol. These results show the importance of the enforcement of the code of conduct.

The assessment of the compliance to the code of conduct revealed very low adherence to the regulations in place in Oslob. Most worrying was the decreasing trend of compliance from 21.4% in 2012 to only 3.4% compliance in 2014 in terms of minimal distance to the whale shark. In comparison, the compliance to a minimum distance of 3 m to the head and 4 m to the tail for whale sharks in Donsol was reported to be 44% in 2005 (Quiros, 2007). Our numbers are conservative because only people within 2 m of the shark were included in the count, whereas the code of conduct regulating the whale shark watching activities in Oslob dictates a minimum distance of 5 m from the side and tail of the sharks, which means that the real compliance might have been even lower. Free swimming, snorkelling guests tended to have lower compliance than guests holding on to the boat while watching the shark underwater. Snorkelers can control the distance to the shark by either actively approaching the animal or swimming away to keep the required distance; nevertheless 85% of snorkelers were too close to the shark in 2014. Boat-holders, in contrast, have little control on the proximity to the shark. The fact that in over three-quarters of the cases boat-holders were too close to the shark in 2014 highlighted the problem that feeders lure the sharks too close to the guest boats. Scuba divers complied with the minimum distance regulation in 74% cases in 2012 and 80% in 2014. Observations further confirmed that crowding around sharks happened regularly. While the ordinance does not specifically state at what distance a guest is considered to interact with the sharks, the maximum number of six people per shark within 10 m of the shark was exceeded in over half of the surveys. There is currently no limit on the number of guests in the interaction area at any one time. Feeders bringing the sharks too close to the tourists could be a result of attempting to improve customer satisfaction. However, while visitors in Ningaloo Reef listed proximity to the whale sharks as a secondary important component of their experience, closeness to the sharks did not affect the overall satisfaction (Davis et al., 1997). Similarly, in whale watching, the actual presence of whales and their behaviour, rather than the proximity of the tour boats to the cetaceans, had an important influence on whale watcher satisfaction (Orams, 2000). In contrast, crowded situations with too many people in the water with physical contact between tourists generally contributed to bad experiences (Davis et al., 1997).

Overall the poor level of compliance to the code of conduct suggests that the tourism pressure on whale sharks in Oslob is very high. Importantly, the observed increase in tolerance levels in experienced whale sharks should not be misinterpreted as a lack of impact (Bejder et al., 2009). While the behavioural response to disturbance was reduced in experienced whale sharks, this does not exclude costs occurring at the physiological level. The costs of group-living in a normally solitary forager were described in detail for the southern stingray (Dasyatis americana; Semeniuk & Rothley, 2008; Semeniuk et al., 2009). Along with increased injuries and ecto-dermal parasites, haematological changes and sub-optimal health were reported for the provisioned animals (Semeniuk et al., 2009). We are currently investigating whether the whale sharks being provisioned at Oslob are affected in a similar way by studying their frequency and accumulation of scars. Araujo et al. (2014) provided evidence for prolonged residency times of provisioned whale sharks in Oslob with unknown consequences on this otherwise highly mobile species. While in natural whale shark aggregations animals increase residency to exploit ephemeral bursts in local productivity until foraging is no longer cost-effective, the food source in Oslob is constant and inexhaustible. At this stage it is still unclear how the previously reported increased residency times in combination with the observed occurrence of conditioning of whale sharks in Oslob may affect the fitness of individual whale sharks. Two possible outcomes can be envisaged: either animals that return regularly fall into an ‘ecological trap’ with the costs of the new habitat outweighing the apparent benefits (Schlaepfer, Runge & Sherman, 2002), or individuals may have identified a predictable, year-round food source that will allow them to grow successfully to maturity without the cost of large-scale movements between other ephemeral foraging areas. Further research should focus on physiological stress levels, body conditions, and diets of provisioned whale sharks in comparison to non-provisioned whale sharks to reveal additional costs and effects on fitness. In terms of recruitment into this aggregation, it would be interesting to study intraspecific differences in stress threshold levels as a predictor for subsequent visitation patterns. It is currently unclear how conditioning in terms of feeding and avoidance patterns may affect the behaviour of whale sharks outside the interaction area. A continued association between boats and food could be detrimental to the whale sharks and lead to increased injuries by boat strikes (Speed et al., 2008). If provisioning had a permanent effect on whale sharks through conditioning, the activities at the study site would no longer fit the initial definition of a non-consumptive wildlife activity as proposed by Duffus & Dearden (1993).

Our results indicated that a reduction in tourism pressure and in reinforcement through provisioning would help to minimise the impact on whale shark behaviour. An improved enforcement of the current code of conduct as well as managing the expectations of the visitors will be a key component in increasing compliance to the ordinance while maintaining customer satisfaction. Additionally, limiting the number of people per boat and the total number of boats in the interaction area would reduce crowding. Feeders should be advised to keep a minimum distance of 5 m to the guest boats. This study highlights the need to consider the past history of a shark when developing codes of conduct, because sharks may show different avoidance behaviours depending on previous experiences with similar stimuli. If possible, codes of conduct based on the precautionary principle should therefore focus on individuals whose behaviour has not been modified through conditioning. While the effects of the observed behavioural modifications on the fitness of the whale sharks remain unclear, the precautionary principle suggests that preventive actions should be taken even if some cause and effect relationships are still unclear (Kriebel et al., 2001).

Overall the results of this study showed that the provisioned whale sharks were capable of adapting their behaviour to a novel food source and increasing their tolerance of potentially disturbing human stimuli. That is why learning leading to behavioural modification is suggested to play an important role in the ecology and feeding plasticity of the whale shark. However, further research is required to test the hypothesis that the initial recruitment of whale sharks to an aggregation is based on chemosensory cues and that the sharks rely on a cognitive spatial map to return to the same feeding site on subsequent occasions.

This work would not have been possible without the help of the Large Marine Vertebrates Project Philippines volunteers (www.lamave.org) and the dedicated work of our staff members, namely Sally Snow, Catherine Lee So, Diana Scalfati, Joseph Murray and Ryan Murray. We’d like to thank the Municipality of Oslob, TOWSFA People’s Organisation, The Department of Environment and Natural Resources—Region 7, The Department of Agriculture—Bureau of Fisheries and Aquatic Resources—Region 7, and Wildbook for Whale Sharks (www.whaleshark.org). A special thank you extends to Dr Jordan Thomson for improving the manuscript and offering advice on statistical analysis.

Additional Information and Declarations

Competing Interests

Author Contributions

Animal Ethics

Field Study Permissions

Data Availability

The authors declare there are no competing interests.

Anna Schleimer conceived and designed the experiments, performed the experiments, analyzed the data, contributed reagents/materials/analysis tools, wrote the paper, prepared figures and/or tables.

Gonzalo Araujo and Alessandro Ponzo conceived and designed the experiments, performed the experiments, analyzed the data, contributed reagents/materials/analysis tools, wrote the paper, prepared figures and/or tables, reviewed drafts of the paper.

Luke Penketh, Anna Heath, Emer McCoy, Jessica Labaja and Anna Lucey conceived and designed the experiments, performed the experiments, contributed reagents/materials/analysis tools, reviewed drafts of the paper.

The following information was supplied relating to ethical approvals (i.e., approving body and any reference numbers):

The whale shark is under the jurisdiction of the Department of Agriculture-Bureau of Fisheries and Aquatic Resources. A Memorandum of Agreement was signed between Physalus NGO and the aforementioned body, and the Department of Environment and Natural Resources-Biodiversity Management Bureau-Region 7.

The following information was supplied relating to field study approvals (i.e., approving body and any reference numbers):

A Gratuitous Permit (GP-01-2013) for Scientific Research was granted by DA-BFAR-7.

A Prior Informed Consent was granted by the Local Community, represented by the Mayor Guaren of Oslob.

The following information was supplied regarding data availability:

Photo-ID data are accessible under Ecocean/Wildbook for Whale Sharks http://www.whaleshark.org.

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
