# Peer review of "Learning from a provisioning site: code of conduct compliance and behaviour of whale sharks in Oslob, Cebu, Philippines"

_PeerJ, doi:10.7717/peerj.1452_

## Round 0.1 · original submission · Minor Revisions

Both reviewers found this to be an interesting and generally well presented piece of research and encouraged publication after some clarifications and additional citations. I am therefore pleased to recommend that it be accepted following minor revisions.

While Reviewer 2 is correct in recognizing that your Discussion is somewhat longer than normal for a print journal, I did not personally find it excessive. I recommend that you carefully consider the content but do not feel that you must remove discussion that you decide is pertinent.

Minor corrections

L269. Replace comma with period
L271. The expression 'full of chemicals' is rather colloquial and not literally correct. I suggest replacing with a more objective phrase.
Table 1. Definition of vertical feeding: Retain singular for consistency ". . . with its body in a vertical orientation with its mouth just below the surface . . . "
Table 1. Definition of dive and eye roll: It is preferable to avoid using the defined term in the definition.
Figure 1 could be more precisely called a frequency distribution than simply a histogram.
Figure 5. For consistency with the other figures, capitalize only the first word of the ordinate.

·

Basic reporting

The Methods section needs improvement; see specific comments below.

Experimental design

No Comments

Validity of the findings

No Comments

Additional comments

- When introducing the topic of effects of shark-based tourism on the focal species (Introduction) as well as when discussing the results (Discussion) the authors have missed the published findings from the Shark Reef Marine Reserve in Fiji. The following papers are relevant here and some of them need to be incorporated:

o Brunnschweiler J.M., Abrantes K.G., Barnett A. (2014) Long-term changes in species composition and relative abundances of sharks at a provisioning site. PLoS ONE 9(1): e86682
o Brunnschweiler J.M., Barnett A. (2013) Opportunistic visitors: long-term behavioural response of bull sharks to food provisioning in Fiji. PLoS ONE 8(3): e58522
o Brunnschweiler J.M., Baensch H. (2011) Seasonal and long-term changes in relative abundance of bull sharks from a tourist shark feeding site in Fiji. PLoS ONE 6(1): e16597
o Brunnschweiler J.M. (2010) The Shark Reef Marine Reserve: a marine tourism project in Fiji involving local communities. Journal of Sustainable Tourism 18: 29-42
o Brunnschweiler J.M., Queiroz N., Sims D.W. (2010) Oceans apart? Short-term movements and behaviour of adult bull sharks Carcharhinus leucas in Atlantic and Pacific Oceans determined from pop-off satellite archival tagging. Journal of Fish Biology: 77: 1343-1358

- In Methods the authors write “Scuba divers currently do not receive the official briefing on the code of conduct and enter the area …” (line 133/134). It appears that scuba divers were pooled with other people in the water in Results (line 251 ff). If this was indeed the case the authors need to explain why they did so. If scuba divers did not get a briefing on the code of conduct how can they know that they should not swim too close to the shark etc.? Also, does “not receive an official briefing” mean that they did receive an unofficial briefing? Please clarify.

- “Compliance with the code of conduct was assessed … within 2 m of the whale shark …” (line 143). Why not using the actual distance of 5 m? And why 2 m and not 3 or 4 m? I note that the authors give an explanation for this (line 146 ff) but I don’t buy the argument “… excludes people that are just marginally breaking the code of conduct.”. You either break it or not. And keeping a distance of 3 or 4 m to the shark is certainly not “marginally” breaking the code of conduct. On line 436 the authors write “… which means that the real compliance might have been even lower.”. Yes certainly, but why not getting closer to the real compliance by using 5 m?

- The Methods section needs considerable clarifications. In particular I had difficulties understanding the paragraph beginning on line 182. For example, it appears that the researchers arrived at the study site before the first feeder boats (line 182). What about the tourists? Do they arrive before or after the feeders? Was everybody (researchers, feeders, tourists) arriving at 6 am? I’m asking because if these sharks are conditioned indeed (associating the presence of boats in the area with food) then one would expect that the sharks turn up independent on what “type” of boat it is (researchers, feeders, tourists). Please clarify. Also, “… time of first shark observed from land were noted.” (line 183/184). Why from land and not directly at the study site? How far away from land is the interaction area of about 65’000 m2 (quite large)? If from land, did the researchers use binoculars to spot the sharks? Please clarify. And a Figure showing the interaction area, land mass etc. would be helpful.

- “… the identification number of the first shark sighted in the water was recorded along with the time …” (line 183/184). This implies that all sharks were identified previously. From Results it appears that also sharks with no identification number turned up. How did the authors assign identification numbers? Please clarify.

- “… trend of becoming new residents, …” (line 241). How is “residency” defined here?

- What I’m missing in Results is a paragraph on the resighting history of individual sharks (n = 59). Form line 294 it appears that 51 sharks were identified since February 2014. Does that mean that during the time before, hence for the majority of the study (31 March 2012 to February 2014), only 8 sharks were “known” individuals (= with an identification number)? Please clarify. Also, was there any seasonality in whale shark sightings/resightings?

- Section 3.2 Compliance (line 250 ff): The authors use the terms “swimmers”, “divers”, “guests”, “snorkelers”. This is confusing; it’s not clear if a swimmer is the same as a snorkeler. I suggest to only use two terms, e.g. “swimmers” and “divers”. This is also related to the comment on pooling divers with swimmers above. I suggest to clearly distinguish between these two groups and do separate analyses since the divers did not get a briefing on the code of conduct.

- “A total of 357 hours of focal follow observations were collected on 59 different sharks.” (line 285). How is this related to the sentence “…, a total of 59 individual sharks were followed, leading to 1109 focal follow observations.” (line 245/246)? The 1109 focal follow observations were of 357 hours duration? And, dividing 357 hours by 59 individual sharks results in 6.05 hours of observation time per shark on average. Please clarify.

- “…, or individuals may have identified … without the cost of long-distance migrations between ephemeral foraging areas.” (line 475/476/477). This is possible but it is equally possible that despite the predictable, year-round food source individual sharks still show long-distance migrations (actually this should be large-scale movements since the term migration refers to the population level), but explore the food source for longer time periods (days to maybe weeks) when in the area; see Brunnschweiler & Barnett 2013 (ref given above) for such an example from bull sharks in Fiji.

- At the end of the paper I would appreciate a short paragraph that specifically addresses the question whether or not provisioning sharks is good or bad.

·

Basic reporting

The article mainly follows all standards of PeerJ. It is well written and clear. My only comment is that the discussion section seems too long and could be shortened to give the reader a clearer discusion of the results presented in this study

Experimental design

First, as mentioned previously, the discussion should be shortened and authors should discuss their finding without going to far in their reflection.
Second, in the methods, it is not clear how GLMM were implemented and if the authors have (or how they have) selected the best model (see model selection methods).

Despite these two comments, globally, the approach and design of the study is interesting and the use of a focal follow framework was interesting and informative. Moreover, the complementary approach linking behavioural modifications of sharks and the compliance of the code of conduct are also interesting as it demonstrates that the code of conduct or its application by skateholders may need to be revised for the activity to remain sustainable.

Validity of the findings

No comments

Additional comments

This paper presents interesting results about the response of whale sharks to a relatively recent provisioning site (the only provisioning activity on a filter-like shark species). The approach is interesting as it uses individual focal follows to quantify the changes in behaviour of sharks that interact with the provisioning activity and also evaluate the compliance of the code of conduct associated with the activity. The authors show that shark’s behaviour has changed along with their experience with the activity and get habituated to the predictable presence of this activity, which suggest some learning capacities. They also showed that the code of conduct is not respected with most tourists being too close to the sharks or even touching them. As the popularity of this touristic industry at this site is growing, rapid changes need to be implemented in the respect of the code of conduct (which may also be adapted). This paper provides evidence of the effect of the activity on sharks themselves as well as some detrimental consequences that may affect the activity over the long-term in terms of sustainability. As some papers that demonstrate that activities are well managed and therefore do not really impact shark’s ecology, paper showing that current applications of code of conduct are not followed properly and require the regulations to be revisited based on evidence of disturbances on shark’s behaviour is important too.
The paper is generally clear and well written and the approach interesting. Dissecting the different behaviours is quite novel and provide a more complete view of the potential changes in shark’s behaviour at the provisioning site. I have however some comments on how the GLMMs were implemented and how the best model was selected. Moreover, I found the paper too long, especially in its discussion section, and I am sure it could be shortened a little and go straight to the important information. In my opinion, this could greatly improve the paper as for the moment I think the reader can get lost with too much information given in the discussion. The authors should concentrate on interpreting their results and not speculating too much.
For these reasons, as the paper is of interest for the readers of PeerJ, I recommend to reconsider the paper after some short revisions mentioned above and below.

Specific comments :
Line 37-42: while only in press at the moment the current paper has been submitted, a review of provisioning on sharks and rays has been published (Brena et al. in press in MEPS) which synthetizes the potential impacts at the individual, group and community scales. If the authors are interested in this review, I will be happy to send them a copy of the manuscript.
Line 48-50 : Maljkovic and Cote 2011 showed also that some individuals were affected in their feeding behaviour
Line 86-99: the link between learning and code of conduct is not clear in this paragraph
Line 141: what do you mean with “six whale shark watchers”? 6 boat or 6 snorkelers?
Line 192-214: in your model, did you try to test the different behaviour as a response variable? How did you select the best GLMM model? It is not mentioned in the text.
Line 245: please change Mar and Jun to Mars and June
Line 300-305: it is not clear if individuals are revisiting the site in consecutive seasons (every year).
Line 318: the end of the sentence seems to be missing (“in these…”)
Line 319-321: how long do sharks stay at the provisioning site?
Line 345-349: it is also the case for other elasmobranch species that were found to arrive at site 1 hour before provisioning start (see Clua et al. 2010 and Gaspar et al. 2008 for example).
Line 349-354: see also Papastamatiou et al. 2011 and Fagan et al. 2013 for spatial map references
Line 354-357: is there any sign of experience? (Larger sharks react faster for example)
Line 358-389: I found this paragraph is too long and should be shortened

References
Brena PF, Mourier J, Planes S, Clua E (in press) Shark and ray provisioning : functional insights into behavioral, ecological and physiological responses across multiple scales. Marine Ecology Progress Series.

Papastamatiou, Y. P., Cartamil, D. P., Lowe, C. G., Meyer, C. G., Wetherbee, B. M., & Holland, K. N. (2011). Scales of orientation, directed walks and movement path structure in sharks. Journal of Animal Ecology, 80(4), 864-874.

Fagan, W. F., Lewis, M. A., Auger‐Méthé, M., Avgar, T., Benhamou, S., Breed, G., ... & Mueller, T. (2013). Spatial memory and animal movement. Ecology letters, 16(10), 1316-1329.

---

## Round 0.2 · accepted · Accept

The changes have improved the manuscript, and I now consider it ready for publication. If you have a chance during production, please replace the comma with a period in 56.1 (L275), replace the A with a in the species name americana (L479) and replace the comma after land with a semicolon (L135).